# Efficient Quasi-2D Perovskite Light-Emitting Diodes Enabled by Regulating Phase Distribution with a Fluorinated Organic Cation

**DOI:** 10.3390/nano12193495

**Published:** 2022-10-06

**Authors:** Ziqing Ye, Junmin Xia, Dengliang Zhang, Xingxing Duan, Zhaohui Xing, Guangrong Jin, Yongqing Cai, Guichuan Xing, Jiangshan Chen, Dongge Ma

**Affiliations:** 1Institute of Polymer Optoelectronic Materials and Devices, State Key Laboratory of Luminescent Materials and Devices, Guangdong Provincial Key Laboratory of Luminescence from Molecular Aggregates, South China University of Technology, Guangzhou 510640, China; 2Joint Key Laboratory of the Ministry of Education, Institute of Applied Physics and Materials Engineering, University of Macau, Macau 999078, China

**Keywords:** perovskite light-emitting diodes, quasi-2D perovskites, fluorinated organic cations, phase distribution, carrier dynamics

## Abstract

Metal halide perovskites have become a research highlight in the optoelectronic field due to their excellent properties. The perovskite light-emitting diodes (PeLEDs) have achieved great improvement in performance in recent years, and the construction of quasi-2D perovskites by incorporating large-size organic cations is an effective strategy for fabricating efficient PeLEDs. Here, we incorporate the fluorine meta-substituted phenethylammonium bromide (*m*-FPEABr) into CsPbBr_3_ to prepare quasi-2D perovskite films for efficient PeLEDs, and study the effect of fluorine substitution on regulating the crystallization kinetics and phase distribution of the quasi-2D perovskites. It is found that *m*-FPEABr allows the transformation of low-n phases to high-n phases during the annealing process, leading to the suppression of n = 1 phase and increasing higher-n phases with improved crystallinity. The rational phase distribution results in the formation of multiple quantum wells (MQWs) in the *m*-FPEABr based films. The carrier dynamics study reveals that the resultant MQWs enable rapid energy funneling from low-n phases to emission centers. As a result, the green PeLEDs achieve a peak external quantum efficiency of 16.66% at the luminance of 1279 cd m^−2^. Our study demonstrates that the fluorinated organic cations would provide a facile and effective approach to developing high-performance PeLEDs.

## 1. Introduction

Metal halide perovskites have attracted great attention in the field of luminescent materials and devices because of their excellent characteristics, such as their high photoluminescence quantum yields (PLQYs), tunable optical band gap, narrower emission spectra, and solution processability [1,2,3,4,5,6,7]. In recent years, the performance of perovskite light-emitting diodes (PeLEDs) has been significantly improved, with external quantum efficiencies (EQEs) surpassing 20% except for blue emission, which shows a promising developmental prospect for applications in lighting and displays [8,9,10,11].

Generally, three-dimensional (3D) metal halide perovskites with the common formula of ABX_3_ possess low exciton binding energy, which is not conducive to radiative recombination and will result in low luminous efficiency [12]. The incorporation of large-size cation groups (noted as L) into ABX_3_ can enable the formation of quasi-two-dimensional (quasi-2D) perovskites with multiphase structure of L_2_(ABX_3_)_n-1_BX_4_, which would induce the effects of dielectric and quantum confinement to increase the exciton binding energy [13,14]. Various methods have been used to enhance the performance of quasi-2D PeLEDs, such as defect passivation, balanced carrier transport, and suppression of Auger recombination [15,16]. In quasi-2D perovskites, rational phase distribution can form effective multiple quantum wells (MQWs), leading to the rapid transfer of exciton energy from lower-n phases to higher-n phases and finally to emitting centers. An efficient energy tunneling can harvest excitons for radiative recombination and thus improve the luminous efficiency. However, unreasonable phase distribution will lead to insufficient energy transfer and energy loss [17,18]. Moreover, quasi-2D perovskites with multiphase structure usually have abundant grain boundaries and surfaces, which easily produce defects that result in trap-assisted recombination [14,19,20,21]. In addition, the formed n = 1 phase, i.e., 2D structure, in quasi-2D perovskites shows strong exciton-phonon coupling at room temperature, which is unfavorable to energy transfer and radiative recombination [13,22,23]. Therefore, it is necessary to accurately regulate phase distribution and effectively reduce defect-induced traps in perovskite films for efficient quasi-2D PeLEDs. The widely used strategies of improving device performance are to combine component engineering of quasi-2D perovskites with anti-solvent treatment, including the introduction of additive or passivation agents [14,23,24,25]. However, the crystallization of quasi-2D perovskites is strongly affected by the dropping parameters during anti-solvent treatment, which is limited by manual manipulation and shows poor reproducibility. Moreover, the additional agents would result in the complicated components of quasi-2D perovskites, which will be not beneficial to mass production of cost-effective PeLEDs in the future.

Phenylethylamine cation (PEA^+^) is the commonly used large-size organic cation for quasi-2D perovskites [26,27,28,29]. There have been many reports on quasi-2D PeLEDs based on PEA^+^ as well as its derivatives [20,30,31,32]. Rational modification of PEA^+^ can enrich its optoelectronic properties and further improve the device performance. The mono-fluorine substitution on the benzene ring of PEA^+^ will not obviously change the ion size, but the strong electron withdrawing feature of the fluorine atom will affect the electron density and dipole moment of the organic cations, leading to a large difference in structural and optoelectronic properties of quasi-2D perovskites [17]. At present, the fluorine-substituted organic cations have been successfully employed for high-performance perovskite solar cells, and only a few studies on their application in PeLEDs were reported [15,17,18,33,34,35,36].

In this study, we demonstrate the efficient green PeLEDs by incorporating the 3-fluorophenylethylammonium bromide (*m*-FPEABr) into CsPbBr_3_ to manipulate the phase distribution of the quasi-2D perovskite films. Compared to the analogue of PEABr, the *m*-FPEABr is found to form rational MQWs in the quasi-2D perovskite films with a spatially more homogeneous distribution of different-n phases, which can harvest the excitons for radiative recombination by efficient energy funneling [12]. In addition, the *m*-FPEABr enables the perovskite films to show larger grains with more ordered orientation, leading to the reduced non-radiative loss caused by the defects at the grain boundaries and interfaces during the energy-funneling process. The champion device achieves the maximum external quantum efficiency (EQE) of 16.66%, which is one of the best EQEs for the green PeLEDs based on the passivation-free quasi-2D perovskites. Our study should provide a facile strategy of incorporating fluorinated organic cations to regulate phase distribution of quasi-2D perovskites for efficient PeLEDs.

## 2. Materials and Methods

### 2.1. Materials

We purchased 3-fluorophenylethylammonium bromide (*m*-FPEABr), phenylethylammonium bromide (PEABr), and poly[bis(4-phenyl)(2,4,6-trimethylphenyl)amine] (PTAA) (average M_W_ = 6000–15,000) from Xi’an Polymer Light Technology Corp. (Xi’an, China). We purchased PbBr_2_ (99.999%), CsBr (99.999%), poly(9-vinylcarbazole) (PVK) (M_W_ = 1,100,000), DMSO (anhydrous, ≥99.9%), ethanol (anhydrous) and chlorobenzene (anhydrous, ≥99.8%) from Sigma-Aldrich (St. Louis, MO, USA). Nickel acetate tetrahydrate (Ni(CH_3_COO)_2_·4H_2_O) (99.9%) was purchased from Shanghai Aladdin Biochemical Technology Co., Ltd. (Shanghai, China). Ethanolamine (99%) was purchased from ACROS, (St. Louis, MO, USA), and 1,3,5-tris(2-N-phenylbenzimidazolyl)-benzene) (TPBI) (≥99%) and LiF (≥99%) were purchased from Jilin OLED Photoelectric Material Corporation (Changchun, China). The molecular structures of *m*-FPEABr and PEABr are shown in Figure 1a. All chemicals were used as received.

### 2.2. Preparation of NiOx and Perovskite Precursor Solutions

The precursor solution of NiOx was prepared according to the previous method [28,37]. Nickel acetate tetrahydrate was dissolved in 10 mL ethanol with 120 µL ethanolamine to get the desired Ni^2+^ concentration of 0.2 M. The solution was stirred under air at 70 °C for 4 h, and then cooled down to room temperature and filtered with a 0.22-μm polytetrafluoroethylene (PTFE) filter.

The precursor solutions of perovskites were prepared by dissolving *m*-FPEABr, PEABr, CsBr, and PbBr_2_ in DMSO at the desired molar ratio in a N_2_-filled glove box. And the concentrations of CsBr and PbBr_2_ were set at 0.3 M. The solutions were stirred at 45 °C overnight, and then was left to stand for 4 h at room temperature and filtered with 0.22 µm PTFE filters before use.

### 2.3. Device Fabrication

The schematic device structure of the PeLEDs is shown in Figure 1b. The devices were fabricated on clean glass substrates precoated with an indium tin oxide (ITO) layer with a thickness of 185 nm. The ITO surface was cleaned with detergents and deionized water by ultrasonication, and then dried at 120 °C for 1 h. The NiO_x_ precursor solution was spin coated on an ITO anode at 4000 r.p.m. and annealed at 350 °C for 60 min in air. The PTAA (8 mg mL^−1^ in chlorobenzene) was spin coated at 4000 r.p.m. and dried at 170 °C for 30 min in a N_2_-filled glove box. Pure chlorobenzene was then spin coated to obtain an ultrathin PTAA layer. After that, the PVK (8 mg mL^−1^ in chlorobenzene) was spin coated on PTAA and then also dried at 170 °C for 30 min. Perovskite films were spin coated at 4000 r.p.m for 120 s and annealed at the desired temperature for 10 min. Finally, the samples were transferred into a thermal evaporator, and the electron-transport layer of TPBi (40 nm), the electron-injection layer of LiF (1 nm), and the cathode of Al (100 nm) were deposited consecutively onto the perovskite film under a high vacuum of less than 4 × 10^−4^ Pa. The active area of the devices is 8 mm^2^, which is determined by the overlap area of Al cathode and ITO anode.

### 2.4. Characterizations

The ultraviolet-visible absorption spectra were measured with a HP 8453 spectrophotometer. The photoluminescence (PL) spectra were measured with a Shimadzu RF-5301PC spectrometer with an excitation wavelength of 365 nm. The PLQYs were measured by a commercialized measurement system XPQY-EQE-350-1100, from Guangzhou Xi Pu Optoelectronics Technology Co., Ltd. with a 365-nm LED as the excitation light source. The time-resolved photoluminescence (TRPL) decay curves of the perovskite films were measured by Edinburgh FLS980 with a 375-nm laser. XRD measurements were performed with a RIGAKU Smart lab (9 kW) X-ray diffractometer with Bragg–-Brentano focusing, a diffracted beam monochromator and a conventional Cu target X-ray tube (λ = 1.5405 Å) set to 40 keV and 30 mA. The grazing incidence wide-angle X-ray scattering (GIWAXS) measurement (incidence angle = 0.3°) was performed on the BL14B1 beamline of the Synchrotron Radiation Facility (SSRF) in Shanghai, China.

The broadband femtosecond transient absorption (TA) spectra of the perovskite films were taken by using the Ultrafast System HARPIA TA spectrometer (Billerica, MA, USA). A Yb: KGW amplifier (PHAROS, Light Conversion, Villnius, Lithuania) supplied laser beams centered at 1030 nm with pulse duration of ~100 fs, pulse repetition rate of 40 kHz, and a maximum pulse energy of 100 µJ. The output of the amplifier was split into two streams of pulses. One was used to drive an optical parametric amplifier (ORPHEUS, Light Conversion) to obtain the pump beam. The residual stream was directed into an ultrafast spectroscopic system (HARPIA-TA, Light Conversion) to generate the white light continuum probe beam. In the spectrometer, the pump chopped at the frequency of 20 kHz was spatially and temporally overlapped with the probe beam on the sample. The pump beam was focused onto the sample with a beam size of 300 µm, which overlapped with the smaller-diameter (200 µm) probe beam. The pump and probe pulses were crossed on the sample's surface, and the transmitted probe beam was recorded by using a CCD linear Si detector coupled to a monochromator. The perovskite samples were placed in a N_2_-filled chamber.

The electroluminescence (EL) characteristics of current density-luminance-voltage (J-L-V) were performed simultaneously by using a computer-controlled source meter (Keithley 2400) equipped with a luminance meter (LS-110, Konica Minolta, Marunouchi, Japan). The EL spectra were recorded with a spectrometer (USB2000+, Ocean Optics, La Jolla, CA, USA). The EQEs were calculated from the luminance, current density, and EL spectra, assuming a Lambertian distribution. All results from the devices were measured in the forward-viewing direction without any out-coupling enhancement techniques. All EL measurements were carried out at room temperature under ambient conditions.

## 3. Results and Discussion

### 3.1. Effect of Organic Cation Content on Performance of Quasi-2D PeLEDs

The large organic cations have a great influence on the structural and optoelectronic properties of quasi-2D perovskites, which can be used to manipulate the performance of quasi-2D PeLEDs. Here, the effect of *m*-FPEABr content on the device performance is first studied. We employ the perovskite precursors of *m*-FPEABr: CsBr: PbBr_2_ with the stoichiometric ratio of *x*: 1: 1 (*x* = 0, 40%, 60%, 80% and 100%) to construct the quasi-2D perovskite films as the emissive layers (EMLs). The perovskite EMLs are prepared by the facile one-step spin-coating method without any anti-solvent treatment, and the annealing temperature is kept at 110 °C. The EL characteristics of the PeLEDs with various *m*-FPEABr contents are shown in Figure 2. For comparison, the PeLEDs based on PEABr are also fabricated by the same procedure, and their EL characteristics are shown in Appendix A. The corresponding parameters of the PeLEDs based on the EMLs with different *m*-FPEABr and PEABr contents are summarized in Appendix A. It can be seen that the PeLED without *m*-FPEABr or PEABr shows the extremely poor EL performance, achieving only a maximum EQE of 0.05% and a maximum luminance of 72 cd m^−2^. The incorporation of large cations can lead to a great improvement in the EL performance and a slight blue shift in the emission peak. The luminance of PEABr-based PeLEDs is increased to 7535 cd m^−2^ at the PEABr content of 40%, but it keeps decreasing with the rise of PEABr content. Moreover, the maximum EQEs of the PEABr-based devices are all below 3%, still showing poor EL performances. As for the PeLEDs based on *m*-FPEABr, the device efficiency is greatly improved from 1.38% to 8.53% as the *m*-FPEABr content is increased from 40% to 60%. When the *m*-FPEABr content reaches 80%, the quasi-2D PeLED achieves a maximum EQE of 11.41% and a maximum luminance of 10,603 cd m^−2^ with the EL peak at 514 nm. Further increasing the *m*-FPEABr content obviously results in the drastic decrease in both EQE and luminance. The above results demonstrate that the incorporation of both PEABr and *m*-FPEABr can improve the device performance, but the maximum luminance will significantly decrease if adding too many organic cations. The decreased luminance would originate from the reduced conductivity of the perovskite films with too much insulating organic cations, which is in agreement with the reduced current density as shown in Figure 2a and Appendix A. Moreover, the low carrier concentration in the EMLs should be unfavorable for radiative recombination. As we can see, the PeLEDs with 80% *m*-FPEABr show the best EL performance; we therefore set the organic cation content as 80% in the precursor solutions to construct quasi-2D perovskite for the subsequent study.

### 3.2. Optimization of Annealing Temperature for Quasi-2D Perovskite EMLs

By using the spin-coating method without anti-solvent treatment to prepare the perovskite EMLs, the crystal growth of perovskite should be strongly dependent on the annealing treatment. Therefore, we next investigate the effect of annealing temperature on the EL characteristics of the quasi-2D perovskites. The device performance of the PeLEDs based on the 80% *m*-FPEABr EMLs annealed at various temperatures are given in Figure 3, and the corresponding parameters are listed in Appendix A. The performance of the devices based on 80% PEABr at different annealing temperatures are also shown in Appendix A and Appendix A for comparison. With the increase of annealing temperature, the two kinds of PeLEDs with *m*-FPEABr and PEABr show a significant upward trend in the maximum luminance; their maximum EQEs increase first and then decrease. It is noteworthy that the champion device based on *m*-FPEABr achieves a high EQE of 16.66% at the annealing temperature of 120 °C, and its maximum luminance is obtained as 10,532 cd m^−2^. When the annealing temperature is increased up to 140 °C, the device based on *m*-FPEABr can realize the maximum luminance of exceeding 28,000 cd m^−2^, even though its maximum EQE is dramatically reduced to 6.55%. As for the PEABr-based devices, a decent EL performance with the maximum EQE of 6.25% can be observed at the annealing temperature of 120 °C, which is still far inferior to that of the champion device based on *m*-FPEABr. These results clearly indicate that the fluorine meta-substitution of PEABr enables the quasi-2D perovskites to significantly improve the EL performance. We believe that the device performance can be further enhanced by optimizing the device structure, such as improving the electrical properties of the transport layers by doping [38].

### 3.3. Phase Distribution of Quasi-2D Perovskite EMLs

From the above investigation of EL devices, it can be seen that the perovskite with 80% *m*-FPEABr shows the optimal EQE at the annealing temperature of 120 °C. To get more insight into the champion quasi-2D perovskite (refer to as *m*-FPEABr perovskite), we then study its optical characteristics, and compare it with the analogue of the PEABr-based quasi-2D perovskite (referred to as PEABr perovskite) which contains 80% PEABr and anneals at the same temperature of 120 °C. Figure 4 show the steady-state absorption and PL spectra of *m*-FPEABr and PEABr perovskites. The absorption peak at 313 nm should be assigned to the zero-dimensional (0D) Cs_4_PbBr_6_, whereas the peaks at 434 nm and 464 nm should be assigned to the n = 2 and n = 3 phases, respectively. There is an ignorable absorption peak at around 400 nm, indicating that few n = 1 phase is formed in the films. The 0D Cs_4_PbBr_6_ can passivate the surface defects of quasi-2D perovskites, and can confine both of carriers and excitons in the EMLs, which are favorable for radiative recombination [28]. Generally, the n = 1 phase can lead to non-radiative loss because of the strong exciton–phonon coupling at room temperature; therefore the suppression of n = 1 phase is conducive to the EL performance of the quasi-2D PeLEDs. Other low-n phases such as n = 2 and 3 can form MQWs with higher-n phases, allowing the energy transfer from the lower-n phases to the higher-n phases and finally harvest to the emissive domains. This kind of energy funneling in MQWs is beneficial for radiative recombination in quasi-2D perovskites. However, the cascade energy transfer strongly depends on the phase distribution in the perovskite EMLs. From the absorption spectra (Figure 4a), it can be seen that the *m*-FPEABr perovskite shows stronger absorption peaks of the n = 2 and 3 phases than the PEABr perovskite, whereas the absorption of the higher-n phases becomes weaker. This indicates that more low-n phases and relatively fewer high-n phases are formed in the *m*-FPEABr perovskite film, which would be more advantageous for energy funneling in the MQWs from the large-bandgap phase of n = 2 to the lowest-bandgap phase for radiative recombination.

From the PL spectra of the two perovskites (Figure 4b), it is obvious that the proportions of their emission from low-n phases are very small, and their PL mainly comes from the high-n phases, which is usually considered to be dominated by the phases with the n values close to ∞ (when n exceeds a certain value, the optical bandgaps of these high-n phases, including the 3D bulk phase, are very close and difficult to distinguish). Meanwhile, the PL intensity of the *m*-FPEABr perovskite from the low-n phases is slightly stronger than that of the PEABr perovskite, which is consistent with its greater composition of low-n phases confirmed by the absorption.

Annealing treatment is an important procedure for perovskite crystal growth, which should affect the phase distribution in the quasi-2D perovskite EMLs. Therefore, we also study the absorption characteristics of the *m*-FPEABr and PEABr perovskites at different annealing temperatures, as shown in Figure 5. In the case of *m*-FPEABr perovskite without annealing, obvious absorption peaks appear at 313 nm, 402 nm, 434 nm, and 464 nm, which indicate that the 0D Cs_4_PbBr_6_ and the low-n phases of n = 1, 2, and 3 can be initially formed during the spin-coating process. However, the absorbance of the high-n phases is very weak, suggesting that the high-n phases were not sufficiently formed before annealing. As the annealing treatment is performed, the absorption peak of the n = 1 phase is significantly decreased, and the peak of the n = 2 phase also shows some reduction, while the absorbance of the high-n phases is gradually enhanced with the annealing temperature. These indicate that the annealing treatment can promote the transform from the low-n phases to the high-n phases, and thus form the rational MQWs in the *m*-FPEABr perovskites, which is beneficial to radiative recombination. For the PEABr perovskite without annealing, there is no obvious absorption peak of 0D Cs_4_PbBr_6_, and the absorption peaks of the low-n phases are very weak, but the absorbance corresponding to the high-n phases are very strong, which indicates that the PEABr perovskite is dominated by the high-n phases before annealing. With an increase in the annealing temperature, the absorption peaks of the n = 2 and 3 phases become distinct, and the peak of the 0D Cs_4_PbBr_6_ obviously emerges, suggesting that the formation of low-dimensional structures is strongly dependent on the annealing process in the PEABr perovskite. From the above analysis, it can be seen that there is a significant difference of the crystallization kinetics between the *m*-FPEABr and PEABr perovskites, which should be caused by the fluorine substitution.

### 3.4. Crystallographic Characteristics of Quasi-2D Perovskite EMLs

The difference of crystallization kinetics can not only lead to the change of phase distribution, but also affect the crystallinity of the quasi-2D perovskites. Consequently, we performed the XRD and GIWAXS measurements to study the crystallographic characterization of the *m*-FPEABr and PEABr perovskites. In the XRD pattern, as shown in Figure 6a, the *m*-FPEABr perovskite at 15.4° (100) and 30.6° (200) and the PEABr perovskite at 15.2° (100) and 30.6° (200) show strong diffraction peaks, corresponding to the CsPbBr_3_ in cubic *Pm-3m* space group. Obviously, the *m*-FPEABr perovskite shows a stronger diffraction peak of the n = 2 phase at 28° than the PEABr perovskite, which is in accordance with the difference in their absorption peaks of the n = 2 phase. We use the Debye–Scherrer equation of D = Kλ/βcosθ to estimate the grain size; here, D is the average thickness of the grain in the direction perpendicular to the crystal plane of the sample, β is the full width at half-maximum (FWHM) of the diffraction peaks in radians, θ is the diffraction angle, λ is the wavelength of the X-ray (Cu Ka: 0.154 nm), and K is the shape factor (taken as 0.89). In order to avoid the interference by the diffraction signal of the n = 2 phase, we choose the peak of (100) plane to calculate the grain size of the high-n phases. The size is calculated to be 15.0 nm in the *m*-FPEABr perovskite, which is larger than 12.8 nm in the PEABr perovskite. Generally, the larger grain size of high-n phases is beneficial to the carrier transport, and it can reduce the non-radiative recombination caused by the grain boundary defects, which is advantageous to improve the performance of PeLEDs.

GIWAXS can be used to probe the crystal orientation in the perovskite films. In Figure 6b,c, the diffraction spots in the region with q_z_ < 10 nm^−1^ should be attributed to the low-n phases. The distinct diffraction spots of the PEABr perovskite at q_z_ = 5.5 nm^−1^ and the *m*-FPEABr perovskite at q_z_ = 8.4 nm^−1^ can be assigned to the (040) and (060) crystal planes of the n = 2 phase, respectively. As can be seen, there are differences in the position and intensity of the n = 2 phase diffraction of two perovskites in the q_z_ direction. These should be caused by the different periodic arrangements of the low-n phases, and also indicate that the introduction of the fluorine atom changes the arrangement of the low-n components. In the region of q > 10 nm^−1^, two perovskites exhibit a combined feature of diffraction rings and spots. The diffraction spots at q = 10.5 nm^−1^, 15.1 nm^−1^, and 21.1 nm^−1^ correspond to the (100), (110), and (200) planes of the 3D CsPbBr_3_-dominated high-n phases, respectively. In the PEABr perovskite, the (100) plane of the 3D dominant phases exhibits strong diffraction spots in both q_z_ and q_xy_ directions, indicating orientation distributions in both in-plane (perpendicular to the substrate) and out-of-plane (parallel to the substrate) directions. Moreover, the (110) plane shows orientation distributions perpendicular to the substrate as well as in the direction with an inclination angle of 45° to the substrate. These results indicate that there exhibit multiple crystallographic orientations of 3D dominant phases in the PEABr perovskite. In contrast, in the *m*-FPEABr perovskite, the (100) plane shows a strong diffraction point in the q_z_ direction, but it shows a significantly weakened signal in the q_xy_ direction, and the diffraction spot corresponding to the (110) plane is also greatly weakened. Meanwhile, the diffraction rings become weaker. These findings suggest that the 3D dominant phases in the *m*-FPEABr perovskite have better crystallinity, and their crystallographic orientation is preferential to the in-plane direction, which are also favorable for carrier transport and device performance.

We also investigate the film morphologies of the *m*-FPEABr and PEABr perovskites by SEM, as shown in Figure 7. Both perovskite films are compact and show high coverage, with small and continuous grains. The PEABr perovskite is flatter and has smaller grain size. In the *m*-FPEABr perovskite, the grain size obviously becomes larger, which confirms the results calculated by the Debye–Scherrer equation according to the XRD measurements.

### 3.5. Carrier Dynamics in Quasi-2D Perovskite EMLs

From the above study, we can see that the *m*-FPEABr perovskite features a more reasonable phase distribution and better crystallinity and crystal orientation than the PEABr perovskite, which will cause the difference in carrier dynamics between the two quasi-2D perovskites. We then carry out the measurements of TRPL and PLQYs to study the recombination characteristics in the two perovskite films, as shown in Figure 8. The TRPL data are fitted with a tri-exponential decay model, the corresponding fitting method and results are shown in Appendix A. It is generally believed that the fast decay process corresponds to the trap-assisted recombination (non-radiative recombination), whereas the slow decay process corresponds to the radiative recombination [25,39,40,41]. It can be observed that the *m*-FPEABr perovskite shows the reduced amplitude constant of fast decay process (*A_1_*) compared with the PEABr perovskite. This indicates that the trap-assisted non-radiative recombination is decreased in the *m*-FPEABr perovskite EML, which is consistent with the lowered defect density due to its larger grain size and better crystallinity. In addition, the *m*-FPEABr perovskite shows the enhanced amplitude constant of slow decay process (*A_3_*), indicating that the proportion of the radiative recombination is increased. The suppression of non-radiative recombination and the promotion of radiative recombination can result in the improvement of the PLQY. From Figure 8b, we can see that the PLQY of *m*-FPEABr perovskite reaches 61.9%, which is much higher than that of 37.5% for the PEABr perovskite. 

In the quasi-2D perovskites, the carrier dynamics is usually affected by the carrier transfer processes in the formed MQWs, which strongly depend on the phase distribution. To study the effect of the phase distribution on the carrier transfer, the TA measurements are performed. The TA results obtained from the two perovskite EMLs are shown in Figure 9. In the PEABr perovskite (Figure 9a–c), there is a strong ground-state bleaching signal at 507 nm, which should be assigned to the high-n phases with n ≥ 5 (the signals overlap so closely that they are indistinguishable). Furthermore, there are only very weak ground state bleaching signals at 409 nm (n = 1), 431 nm (n = 2), and 461 nm (n = 3), which coincides with the low content of the low-n phases, as discussed above. On the other hand, the bleaching signals of the low-n phases could be weakened by the interference of the positive signals of the nearby excited state absorption. In contrast, in the *m*-FPEABr perovskite (Figure 9d–f), distinct ground-state bleaching peaks can be observed at 430 nm (n = 2), 460 nm (n = 3), 477 nm (n = 4), and 503 nm (n ≥ 5), respectively, and a very faint bleaching signal is present at 406 nm (n = 1). It is easy to see that the TA spectra of the two perovskites are in general agreement with the steady-state absorption results (Figure 4a).

In general, the intensity evolution of the ground-state bleaching peak corresponds to the concentration change of photogenerated carriers in the correlative phases. The variation of carriers in each phase is related to the carrier transfer and recombination processes, which vary in the time scales. To explore the carrier transfer process, the decay kinetics of the ground-state bleaching at the selected wavelengths are fitted (Figure 9c,f); the method and the results are shown in Appendix A [14,39,42]. In the *m*-FPEABr perovskite, the low-n phases with n = 2, 3, and 4 show the ultrafast decay times (*τ_1_*) of 0.23 ps, 0.31 ps, and 0.30 ps, respectively. For the high-n phases with n ≥ 5, the bleaching peak formation time (*τ_r_*) is 0.38 ps. The ultrafast decay of the bleaching peaks of n = 2, 3, and 4 phases and the formation of the bleaching peaks of n ≥ 5 phases are essentially in the same time range, indicating the existence of the efficient carrier transfer from the low-n phases to the high-n phases. The ultrafast carrier transfer in the MQWs of the *m*-FPEABr perovskite should be conducive to effectively harvest exciton energy for emission. However, the carrier transfer in the PEABr perovskite can be neglected because the contents of low-n phases are too low to form the effective structure of MQWs. 

From the above discussion, we can clearly see that the phase distribution has a significant effect on the carrier dynamics of the quasi-2D perovskites. Remarkably, the difference in the phase distribution originates from the introduction of a single fluorine atom on the benzene ring of phenylethylamine cation. This molecular structure perturbation should affect the intermolecular interactions. For the purpose of studying the effect of the molecular structure of organic cations on the assembly of perovskite structure, we estimated the formation enthalpy (Δ*H_f_*) of the n = 1 phase of the two perovskites at T = 0 K with density functional theory (DFT), and the calculation parameters and results are shown in Appendix A [17,43,44,45,46]. Although the two quasi-2D perovskites contain multiple phases, we only chose the typical n = 1 phase for the theoretical calculation. The reason is that the n = 1 phase is only composed of the basic unit of [PbBr_6_]^4−^ octahedra and large organic cation, which allows us to focus on the interaction between inorganic octahedra and organic cations [17]. The Δ*H_f_*
^0K^ is determined to be −0.76 eV per unit for *m*-FPEA_2_PbBr_4_ and −0.48 eV per unit for PEA_2_PbBr_4_. This suggests that *m*-FPEABr is much easier than PEABr to form the 2D structure, which is consistent with the absorption results of the quasi-2D perovskites before annealing (Figure 5). Accordingly, it is understandable that the other low-n phases, such as n = 2, 3, and 4, are more easily formed in the *m*-FPEABr perovskite during the spin coating. After annealing treatment, the low-n phases in the *m*-FPEABr perovskite are gradually decreased, especially for the n = 1 phase, which can be attributed to the transformation of the low-n phases to the high-n phases. 

## 4. Conclusions

We employed a fluorinated organic cation to prepare quasi-2D perovskite films with an MQW structure by simple one-step spin coating without anti-solvent treatment for efficient green PeLEDs. The combination of *m*-FPEABr and CsPbBr_3_ was demonstrated to form the quasi-2D perovskites with homogeneous phase distribution, which enable efficient energy funneling from low-n phases to high-n phases in the MQWs. By contrast, the control quasi-2D perovskites based on the widely used PEABr presented inhomogeneous phase distribution with significantly fewer low-n phases. More interestingly, the high-n phases in the *m*-FPEABr-based quasi-2D perovskites displayed more ordered crystal orientation preferentially among the out-of-plane direction. Additionally, the grain sizes became larger when using *m*-FPEABr to replace PEABr in the quasi-2D perovskite films, which can lower the effect of defects on the energy funneling and carrier transporting. By manipulating the phase distribution and crystallinity of the quasi-2D perovskites by *m*-FPEABr, the green PeLEDs achieved a peak EQE of 16.66%, which is among the best EL performances of the perovskites without anti-solvent treatment or addition of a passivation agent. Our study should shed light on the rational regulation of crystallization kinetics to realize efficient energy funneling in quasi-2D perovskite films for high-performance PeLEDs.

## Figures and Tables

**Figure 1 nanomaterials-12-03495-f001:**
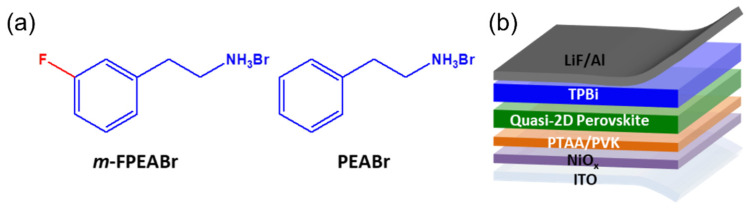
(**a**) Molecular structures of *m*-FPEABr and PEABr; (**b**) Schematic diagram of device structure of perovskite light-emitting diodes (PeLEDs).

**Figure 2 nanomaterials-12-03495-f002:**
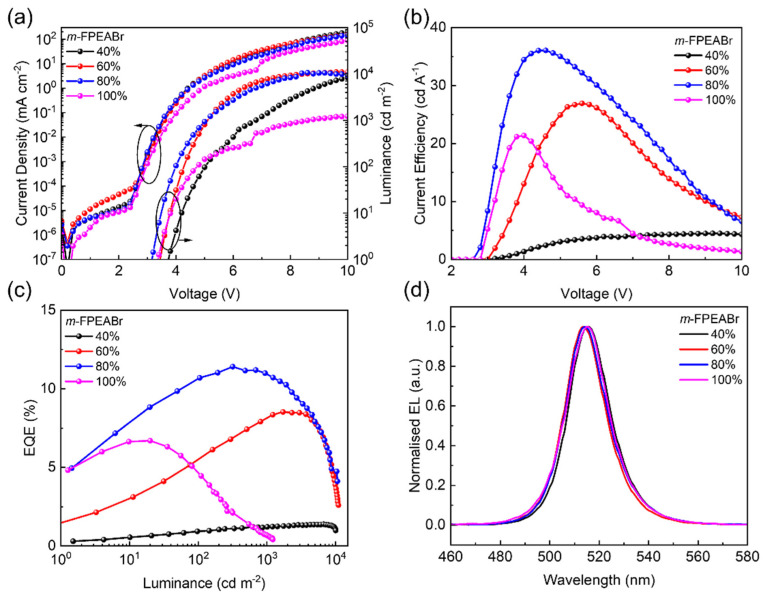
Performance of quasi-2D PeLEDs based on different contents of *m*-FPEABr at 110 °C annealing temperature. (**a**) Current density-voltage-luminance (J-V-L) characteristics. (**b**) Current efficiency-voltage (CE-V) characteristics. (**c**) EQE-luminance (EQE-L) characteristics. (**d**) EL spectra at 6V.

**Figure 3 nanomaterials-12-03495-f003:**
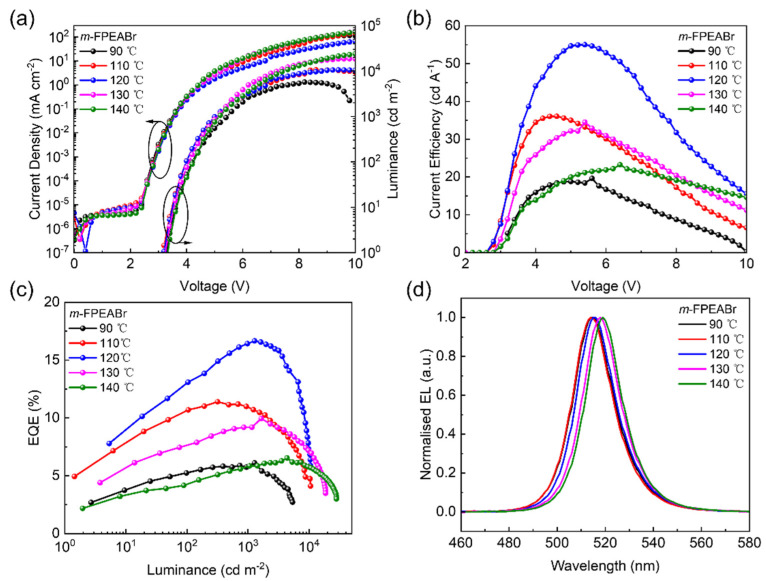
Performance of quasi-2D PeLEDs based on 80% *m*-FPEABr at different annealing temperatures. (**a**) C-V-L characteristics. (**b**) CE-V characteristics. (**c**) EQE-L characteristics. (**d**) EL spectra at 6V bias voltage.

**Figure 4 nanomaterials-12-03495-f004:**
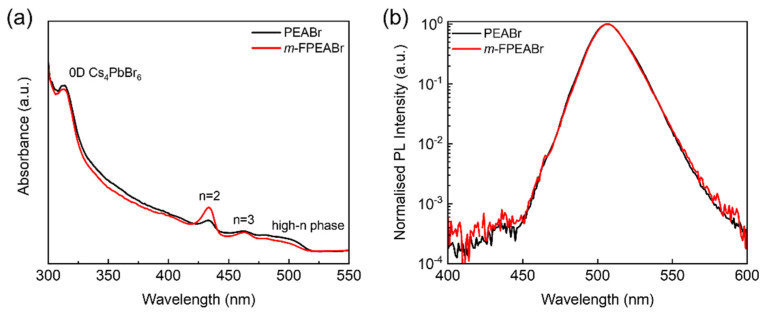
(**a**) Absorption and (**b**) PL spectra of *m*-FPEABr and PEABr perovskite films.

**Figure 5 nanomaterials-12-03495-f005:**
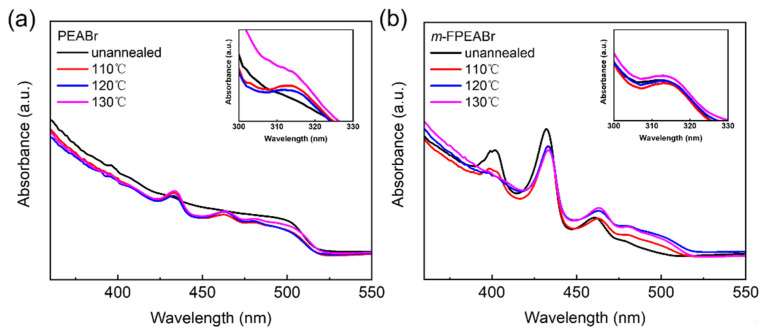
Absorption spectra of (**a**) PEABr and (**b**) *m*-FPEABr perovskite films at different annealing temperatures.

**Figure 6 nanomaterials-12-03495-f006:**
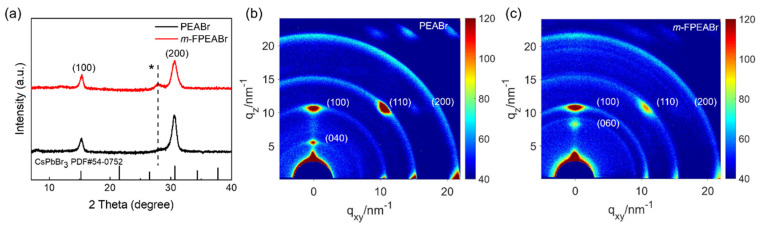
(**a**) XRD patterns of PEABr and *m*-FPEABr perovskite films. GIWAXS patterns of (**b**) PEABr and (**c**) *m*-FPEABr perovskite films. *: low-n phase.

**Figure 7 nanomaterials-12-03495-f007:**
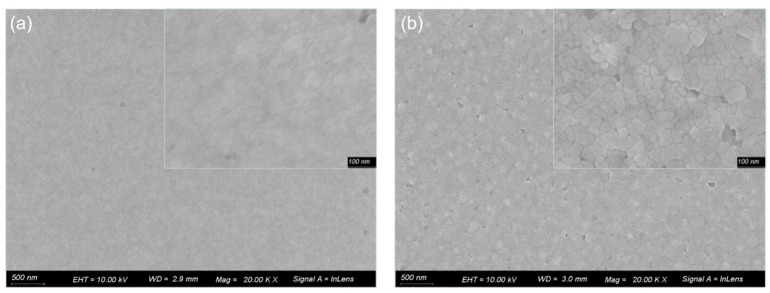
SEM images of (**a**) PEABr and (**b**) *m*-FPEABr perovskite films.

**Figure 8 nanomaterials-12-03495-f008:**
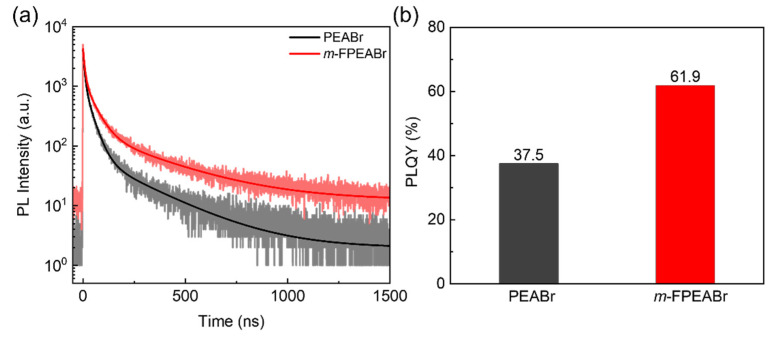
(**a**) TRPL spectra and (**b**) PLQYs of PEABr and *m*-FPEABr perovskite films.

**Figure 9 nanomaterials-12-03495-f009:**
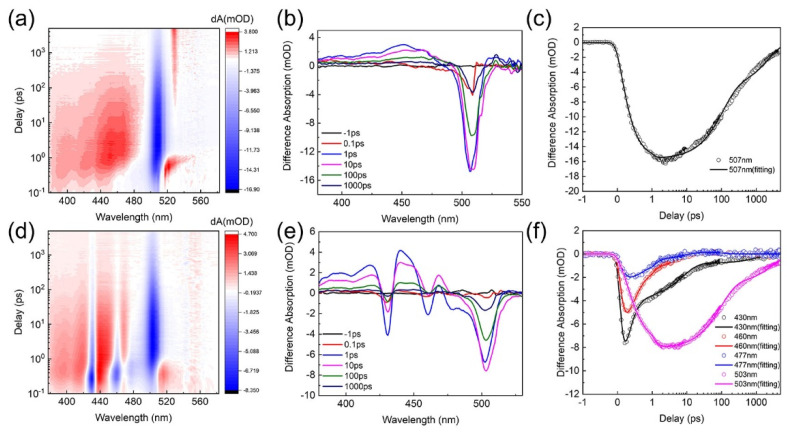
(**a**) Decay-wavelength-absorption characteristics. (**b**) Decay kinetics at different probe times. (**c**) Decay kinetics at different wavelengths of PEABr perovskite. (**d**) Decay-wavelength-absorption characteristics. (**e**) Decay kinetics at different probe times. (**f**) Decay kinetics at different wavelengths of *m*-PEABr perovskite.

## Data Availability

The data that support the findings of this study are available from the corresponding author upon reasonable request.

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
