# Peer review of "Efficient Quasi-2D Perovskite Light-Emitting Diodes Enabled by Regulating Phase Distribution with a Fluorinated Organic Cation"

_nanomaterials, 2022, doi:10.3390/nano12193495_

Round 1
Reviewer 1 Report
In this work, the authors presented the construction of quasi-2D perovskites by incorporating large-size organic cations, fluorine meta-substituted phenethylammonium bromide (m-FPEABr), for fabricating efficient PeLEDs. A peak EQE of 16.66% of the green PeLEDs was achieved by manipulating the phase distribution and crystallinity of the quasi-2D perovskites by m-FPEABr. This study is interesting and helpful for improving the performance of PeLEDs. However, there are some points should be addressed:
l Could the authors explain what is “energy funneling” which appeared in this article several times? (Line 27, 84, 87…)
l The authors indicated that the relatively high preferential orientation of m-FPEABr perovskite is one of the reasons for its superior property based on the GIWAXS results. However, I can’t follow the discussions on lines 336-348. For example: the result of a strong (110) diffraction spot is attributed to the less preferred orientation in PEABr perovskite. But, a cubic with the (100) plane parallel to the substrate means the (110) plane is in the direction with an inclination angle of 45° to the substrate. Could the authors explained more clearly why the GIWAXS results indicate that there exhibit multiple crystallographic orientations of 3D dominant phases in the PEABr perovskite.
l The TRPL data are fitted with a tri-exponential decay model, the corresponding fitting method and results are shown in Table S3. The authors said that the fast decay process corresponds to trap-assisted recombination (non-radiative recombination), while the slow decay process corresponds to bimolecular recombination (radiative recombination). It is contrary to my knowledge that the fast decay process is caused by bimolecular recombination of photo-generated free carriers, whereas the slow decay process is attributed mainly to trap-assisted recombination. I suggest the authors to refer to the reference papers about the discussions of TRPL.
Reviewer 2 Report
Author compare m-FPEABr and PEABr these two molecules blended in CsPb/PbBr2 precursor solution for fabricating PeLEDs device.
1. Blending fluorinated cation is interesting. I suggest author could review and cite Lin et al. (RSC Advances 2018, 8, 12526-12534), which focus on adding Bphene in PC61BM cluster for increasing solar cell efficiency.
2. The ratio (0~100%) of m-FPEABr or PEABr to CsBr and PbBr2. The ratio is determined by molar concentration? or weight concentration ?
3. Is possible to quantitatively characterize how much n=1, n=2, n=3, each phases in provskite emission layer ?
4. Why Fig 4b shows identical PL for m-FPEABr and PEABr ?
Reviewer 3 Report
In this paper, authors have used fluorine substituted m-FPEABr to prepare quasi-2d Perovskite LED and achieved good efficiency of 16.6%. Authors has also tried to explain the mechanism for improvement in the efficiency compared with the PEABr. The study will help to identify and further develop alternative cations to fabricate the quasi-2d perovskite LEDs. Reviewer supports the publication of the paper with following recommendations.
- The stability is an important matrix for perovskite LEDs, authors should provide stability data of their LEDs as their devices shows strong efficiency roll off.
- Since high efficiencies were reported previously for the quasi-2d Perovskite LED by different cations and process, authors should compare their results with previous results and discuss the advantage of using their method.
https://www.nature.com/articles/s41467-020-20555-9
https://onlinelibrary.wiley.com/doi/full/10.1002/adom.201900747
Round 2
Reviewer 1 Report
This article has been improved. I recommend it for publication.